# Instance Correlation Graph-based Naive Bayes

**Chengyuan Li** [1]  **Liangxiao Jiang** [1]  **Wenjun Zhang** [1]  **Liangjun Yu** [2]  **Huan Zhang** [3]

## Abstract

Due to its simplicity, effectiveness and robustness, naive Bayes (NB) has continued to be one of the top 10 data mining algorithms. To improve its performance, a large number of improved algorithms have been proposed in the last few decades. However, in addition to Gaussian naive Bayes (GNB), there is little work on numerical attributes. At the same time, none of them takes into account the correlations among instances. To fill this gap, we propose a novel algorithm called instance correlation graph-based naive Bayes (ICGNB). Specifically, it first uses original attributes to construct an instance correlation graph (ICG) to represent the correlations among instances. Then, it employs a variational graph auto-encoder (VGAE) to generate new attributes from the constructed ICG and uses them to augment original attributes. Finally, it weights each augmented attribute to alleviate the attribute redundancy and builds GNB on the weighted attributes. The experimental results on tens of datasets show that ICGNB significantly outperforms its deserved competitors. Our codes and datasets are available at https://github.com/jiangliangxiao/ICGNB.

## 1. Introduction

Supervised classification is one of the most fundamental and significant tasks in data mining (Han et al., 2011). Due to the explicit interpretability and the powerful expression ability, Bayesian networks (Pearl, 1989; Friedman et al., 1997; Tang et al., 2016; Zhang et al., 2020a) are commonly used in supervised classification. Assume that $A_1, A_2, ..., A_j, ..., A_m$ are $m$ attribute variables, an instance $\boldsymbol{x}$ can be represented as

an attribute value vector $< a_1, a_2, ..., a_j, ..., a_m >$, where $a_j$ is the value of $\boldsymbol{x}$ on $A_j$. Let $C$ represent the class variable and $c$ represent the value that $C$ takes, Bayesian networks use Eq. (1) to classify $\boldsymbol{x}$:

$$\hat{c}(\boldsymbol{x}) = \underset{c \in C}{arg\,max}\, \pi_c P(a_1, a_2, ..., a_j, ..., a_m | c), \quad (1)$$

where $\hat{c}(\boldsymbol{x})$ is the class label of $\boldsymbol{x}$ predicted by Bayesian networks and $\pi_c$ is the prior probability of $c$. Because directly estimating $P(a_1, a_2, ..., a_j, ..., a_m | c)$ is an NP-hard problem (Chickering, 1995), naive Bayes (NB) proposes an assumption that all attributes are fully independent given the class, i.e. attribute conditional independence assumption. Based on this assumption, NB uses Eq. (2) to classify $\boldsymbol{x}$:

$$\hat{c}(\boldsymbol{x}) = \underset{c \in C}{arg\,max}\, \pi_c \prod_{j=1}^{m} \theta_{a_j|c}, \quad (2)$$

where $\theta_{a_j|c}$ is the conditional probability of $a_j$ given $c$.

Despite its simplicity, NB has demonstrated remarkable performance and continued to be one of the top 10 data mining algorithms (Wu et al., 2008). Nevertheless, the attribute conditional independence assumption is difficult to hold in reality, which limits the performance of NB. To mitigate the attribute conditional independence assumption, numerous improved algorithms of NB have been proposed, which can be broadly divided into four categories: structure-oriented, probability-oriented, attribute-oriented and instance-oriented algorithms.

However, in addition to Gaussian naive Bayes (GNB), there is little work on numerical attributes. At the same time, none of them takes into account the correlations among instances. To address these issues, we propose a novel algorithm called instance correlation graph-based naive Bayes (ICGNB). Firstly, we construct an instance correlation graph (ICG) using original attributes to represent the correlations among instances. Then, we employ a variational graph auto-encoder (VGAE) (Kipf & Welling, 2016) to generate new attributes from the constructed ICG and use them to augment original attributes. Finally, we weight each augmented attribute by maximizing the model's conditional log-likelihood (CLL) to alleviate the attribute redundancy and then build GNB on the weighted attributes. In summary, the contributions of this paper can be highlighted as:

---

[1]School of Computer Science, China University of Geosciences, Wuhan 430074, China [2]College of Computer, Hubei University of Education, Wuhan 430074, China [3]School of Computer Science and Artificial Intelligence, Zhengzhou University, Zhengzhou 450001, China. Correspondence to: Liangxiao Jiang <ljiang@cug.edu.cn>.

*Proceedings of the 42ⁿᵈ International Conference on Machine Learning*, Vancouver, Canada. PMLR 267, 2025. Copyright 2025 by the author(s).

- We argue that the correlations among instances should be introduced to improve NB. In contrast to existing improved algorithms focusing on individual instances, we intend to mine additional information across instances.

- We develop a novel instance correlation graph (ICG)-based representation learning method, which exploits the correlations among instances to generate new attributes and thus enhances the identification abilities of the original attributes.

- We propose an instance correlation graph-based naive Bayes (ICGNB) algorithm, which enjoys the advantages of attribute generation, attribute augmentation and attribute weighting and thus provides a new pathway to improve NB.

The rest of this paper is organized as follows: Section 2 conducts a survey on improved algorithms of NB. Section 3 describes the proposed ICGNB in detail. Section 4 reports the experiments and results. Section 5 concludes this paper and outlines the research directions for future work.

## 2. Related work

In recent years, numerous improved algorithms of NB have been proposed, which can be broadly divided into four categories: structure-oriented, probability-oriented, attribute-oriented and instance-oriented algorithms.

Structure-oriented algorithms (Webb et al., 2005; Jiang et al., 2009; Qiu et al., 2015) improve NB by extending its network structure. Specifically, the original network structure lacks edge connections among attribute vertices, which restricts the ability of NB to represent attribute dependencies. To overcome this limitation, structure-oriented algorithms extend the network structure by adding directed arcs, which point from the parent vertices to the child vertices to represent attribute dependencies. Distinguished from NB, where the conditional probabilities are calculated given the class vertex, in structure-oriented algorithms, the conditional probabilities of each attribute are calculated given both the class vertex and its parent vertices.

Probability-oriented algorithms (Hindi, 2014; Diab & Hindi, 2017; Hindi et al., 2020; Zhang & Jiang, 2022) provide a strategy to obtain more reliable conditional probabilities than those estimated by NB. In NB, conditional probabilities are roughly estimated based on frequencies, however, they might be unreliable when a dataset is too small or the attribute conditional independence assumption is violated. Probability-oriented algorithms adjust them to be more reliable conditional probabilities by employing eager and lazy learning. Eager learning spends the main calculating cost in the training stage, while lazy learning spends the main calculating cost in the classification stage.

Attribute-oriented algorithms handle attributes through different strategies, which can be further divided into attribute selection, attribute weighting and attribute generation. In attribute selection (Hall, 2000; Tang et al., 2016; Chen et al., 2020), irrelevant and redundant attributes are removed from original attributes to obtain an optimal subset of attributes. Attribute selection includes filters and wrappers, which remove attributes by employing general data characteristics and classification accuracies of NB, respectively. In attribute weighting (Zaidi et al., 2013; Jiang et al., 2019b; Zhang et al., 2020b), due to attributes holding different importance on classification, they should be assigned different weights based on their predictive abilities (Lee et al., 2011). Evaluating the predictive ability of each attribute is the most crucial problem of attribute weighting. The same as attribute selection, attribute weighting also includes filters and wrappers. In attribute generation (Ou et al., 2022; He et al., 2023; Zhang et al., 2023), new attributes are generated by mapping original attributes to another attribute space, which can capture latent characteristics of the data. New attributes can enhance the identification abilities of original attributes by replacing, augmenting or acting as a new view.

Instance-oriented algorithms can also be further divided into instance selection, instance weighting and instance generation. In instance selection (Bilmes & Ng, 2009; Langley & Sage, 2013; Wang et al., 2015), a local subset of instances is selected to build NB. Considering that the attribute conditional independence assumption is difficult to satisfy on the entire training instances, however, it may hold by selecting a local subset of instances for a given test instance. In instance weighting (Jiang et al., 2012; 2014; Xu et al., 2019), each instance is assigned a numerical weight between 0 and 1 based on its reliability. Then the weight of each instance is incorporated into the formulae of the prior and conditional probabilities to get a more accurate probability estimation. In instance generation (Jiang et al., 2005; Jiang & Zhang, 2005; Jiang et al., 2008), new instances are generated based on the original instances and added to the dataset. The instance distribution of the data is optimized by using new instances and original instances simultaneously.

To the best of our knowledge, except for two attribute-oriented algorithms (Ou et al., 2022; He et al., 2023), almost all above improved algorithms focus on nominal attributes. Moreover, all of these improved algorithms consider individual instances solely and ignore the correlations among instances. These limitations motivate us to propose a more comprehensive algorithm that can handle numerical attributes and leverage the correlations among instances.

## 3. ICGNB

To address the limitations mentioned above, we first develop a novel ICG-based representation learning method. In our

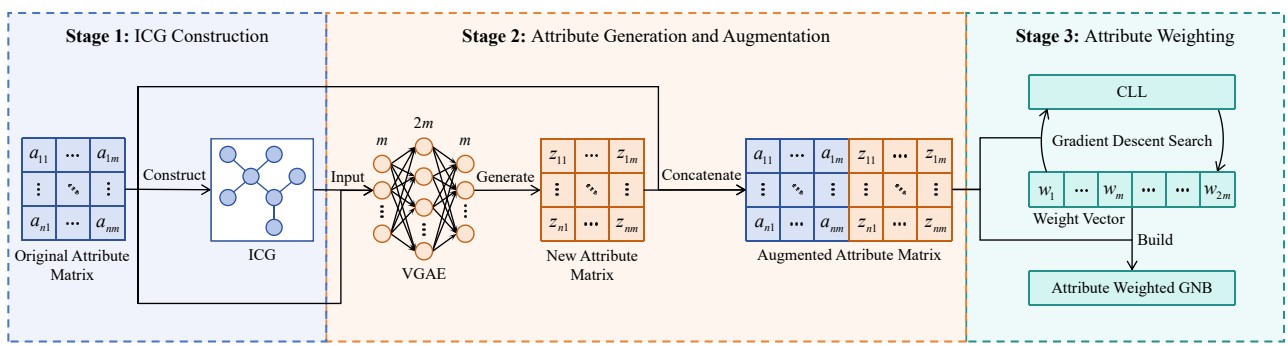

Figure 1. Framework of ICGNB.

method, we construct an ICG to represent the correlations among instances and transform the correlations into new attributes to augment original attributes. Based on the developed representation learning method, we further propose a novel algorithm called instance correlation graph-based naive Bayes (ICGNB). The framework of ICGNB is graphed in Figure 1. From Figure 1, we can see clearly that ICGNB is a three-stage algorithm. In the first stage, we mine the correlations among instances from the original attribute matrix and construct an ICG to represent the correlations. In the second stage, we input ICG and the original attribute matrix into VGAE to generate a new attribute matrix and augment the original attribute matrix by it. In the third stage, we maximize the CLL by the gradient descent search to optimize the weight vector and finally build attribute weighted GNB on augmented attributes.

### 3.1. ICG construction

In this subsection, we construct ICG to connect all instances into a complete structure and represent the correlations as an undirected graph. ICG presents a topological structure of the data, connecting instances with high correlations while separating those with low correlations. In ICG, to avoid containing redundant correlations, we construct it as a sparse graph. Under the condition that all instances should be connected into a complete structure, we add as few edges as possible to ICG.

Given a dataset with $n$ instances and $m$ attributes, it can be represented as $\mathcal{D} = \{X, c\}$, where $X \in \mathbb{R}^{n \times m}$ is the original attribute matrix, $c$ is the class label vector. The $i$-th row in $X$ is represented as $x_i$, corresponding to the $i$-th instance. The ICG is represented as $O = <V, E>$, where $V$ is the set of vertices, and $E$ is the set of edges. Each vertex in ICG represents an instance, and each edge represents a connection between two instances. In addition, we define $E_F$ as the set of edges in a full connection graph of instances, which contains $n(n-1)/2$ edges. The ICG is constructed using original attributes, and the construction (ICG-construction) process is depicted by **Algorithm 1**.

---

**Algorithm 1** ICG-construction($X$)

1: **Input:** $X$ - the original attribute matrix.
2: **Output:** $E$ - the set of edges in ICG.
3: Construct a full connection graph of instances and store its edges in $E_F$;
4: **for** $i = 1$ to $n$ **do**
5:     **for** $t = 1$ to $n$ **do**
6:         Calculate $d(x_i, x_t)$ between $x_i$ and $x_t$ by Eq. (3);
7:     **end for**
8: **end for**
9: Sort edges in $E_F$ by Euclidean distances in ascending order;
10: Initialize an empty set $E$;
11: **for** $i = 1$ to $n(n-1)/2$ **do**
12:     **if** two vertices connected by the $i$-th edge in $E_F$ are not reachable through the edges in $E$ **then**
13:         Add the the $i$-th edge in $E_F$ to $E$;
14:         **if** $E$ contains $n$-1 edges **then**
15:             Break;
16:         **end if**
17:     **end if**
18: **end for**
19: **for** $i = 1$ to $n$ **do**
20:     Add a self-connecting edge for the $i$-th vertex to $E$;
21: **end for**
22: **return** $E$.

---

Firstly, we construct a full connection graph of instances and store its edges in $E_F$. Then, we calculate the distances between instances two by two and sort edges in $E_F$ by the distances in ascending order. In this paper, we take advantage of the widely used Euclidean distance to measure the distance $d(x_i, x_t)$ between two instances $x_i$ and $x_t$, which can be formulated as Eq. (3):

$$d(x_i, x_t) = \sqrt{\sum_{j=1}^{m} (a_{ij} - a_{tj})^2}, \qquad (3)$$

where $a_{ij}$ and $a_{tj}$ are the $j$-th attribute values of $x_i$ and $x_t$, respectively. Next, we process each edge in $E_F$ iteratively, checking whether the two vertices connected by it are reachable through the edges in $E$. If not reachable, the edge is added to $E$. This process continues until exactly $n$-1 edges have been added. Finally, we add a self-connecting edge

for each vertex to consider the correlation between each instance and itself. After constructing ICG, we transform $O$ into an adjacency matrix $\boldsymbol{G}$. The element of $\boldsymbol{G}$ in the $i$-th row and the $j$-th column is represented as $G_{ij}$, which takes the value 1 if there is an edge between the $i$-th vertex and the $j$-th vertex and 0 otherwise.

### 3.2. Attribute generation and augmentation

In this subsection, to fully leverage the correlations among instances and mine additional information across instances, we take an embedding matrix generated from ICG as the new attribute matrix for instances. During the generation of the embedding matrix, the graph convolution operation in VGAE plays a crucial role, which calculates the response of a vertex based on its neighbors through Fourier transform(Kipf & Welling, 2017). Through the graph convolution operation, we encode the correlations among each instance and its neighbors into the embedding matrix.

In VGAE, the embedding matrix $\boldsymbol{Z}$ consists of $n$ embedding vectors, and the $i$-th row in $\boldsymbol{Z}$ is the $i$-th embedding vector $\boldsymbol{z}_i$. Similar to the variational auto-encoder (VAE) (Kingma & Welling, 2014), VGAE contains an encoder $q(\boldsymbol{Z}|\boldsymbol{X}, \boldsymbol{G})$, a decoder $p(\boldsymbol{G}|\boldsymbol{Z})$ and a variational prior $p(\boldsymbol{Z})$. Among them, the encoder $q(\boldsymbol{Z}|\boldsymbol{X}, \boldsymbol{G})$ transforms $\boldsymbol{X}$ and $\boldsymbol{G}$ into $\boldsymbol{Z}$ based on $p(\boldsymbol{Z})$. The decoder $p(\boldsymbol{G}|\boldsymbol{Z})$ reconstructs $\boldsymbol{G}$ from $\boldsymbol{Z}$. According to the mean field theory (Blei et al., 2016), $p(\boldsymbol{Z})$ can be expressed by several mutually independent components. Specifically, treating the embedding vectors as the independent components, the process of generating $\boldsymbol{Z}$ through $q(\boldsymbol{Z}|\boldsymbol{X}, \boldsymbol{G})$ can be represented as Eq. (4):

$$q(\boldsymbol{Z}|\boldsymbol{X}, \boldsymbol{G}) = \prod_{i=1}^{n} \mathcal{M}(\boldsymbol{z}_i|\phi_i), \qquad (4)$$

where $\mathcal{M}(\boldsymbol{z}_i|\phi_i)$ is the vector prior of $\boldsymbol{z}_i$, $\phi_i$ is the parameter set of $\mathcal{M}(\boldsymbol{z}_i|\phi_i)$. For each embedding vector, we design a mixture of two multivariate Gaussian distributions with diagonal covariance as the vector prior (Blundell et al., 2015). The two multivariate Gaussian distributions have the same means but different variances, and the vector prior can be represented as Eq. (5):

$$\begin{aligned} &\mathcal{M}(\boldsymbol{z}_i|\phi_i) \\ =&\mathcal{M}(\boldsymbol{z}_i|\boldsymbol{\mu}_i, \boldsymbol{\sigma}_{1i}, \boldsymbol{\sigma}_{2i}) \\ =&\frac{1}{2}(\mathcal{N}(\boldsymbol{z}_i|\boldsymbol{\mu}_i, diag(\boldsymbol{\sigma}_{1i}^2)) + \mathcal{N}(\boldsymbol{z}_i|\boldsymbol{\mu}_i, diag(\boldsymbol{\sigma}_{2i}^2))), \end{aligned} \qquad (5)$$

where $\boldsymbol{\mu}_i$ is the mean vector, $\boldsymbol{\sigma}_{1i}^2$ and $\boldsymbol{\sigma}_{2i}^2$ are the variance vectors, $diag(\boldsymbol{\sigma}_{1i}^2)$ and $diag(\boldsymbol{\sigma}_{2i}^2)$ are the diagonal covariance matrices. The variances of the first multivariate Gaussian distribution are configured to be considerably larger than those of the second, i.e. $\boldsymbol{\sigma}_{1i}^2 >> \boldsymbol{\sigma}_{2i}^2$. $\boldsymbol{\sigma}_{1i}^2$ and $\boldsymbol{\sigma}_{2i}^2$ increase the diversity of the vector prior by providing a heavier tail and a higher peak in the two multivariate Gaussian

distributions, respectively. $\boldsymbol{\mu}_i$, $\boldsymbol{\sigma}_{1i}$ and $\boldsymbol{\sigma}_{2i}$ are inferred by the graph convolution operation. Formally, the graph convolution operation from the $r$-th layer to the $(r+1)$-th layer in VGAE can be represented as Eq. (6):

$$\boldsymbol{H}^{r+1} = \delta(\tilde{\boldsymbol{D}}^{-\frac{1}{2}} \boldsymbol{G} \tilde{\boldsymbol{D}}^{-\frac{1}{2}} \boldsymbol{H}^r \boldsymbol{W}^r), \qquad (6)$$

where $\delta(\cdot)$ is a RELU function, $\tilde{\boldsymbol{D}}$ is the degree matrix of $\boldsymbol{G}$, $\tilde{D}_{ii} = \sum_{j=1}^{n} G_{ij}$, $\boldsymbol{W}^r$ is the weight matrix in the $r$-th layer, $\boldsymbol{H}^r$ is the activation matrix in the $r$-th layer, $\boldsymbol{H}^0 = \boldsymbol{X}$. The inference of means and variances can be represented as Eq. (7) and Eq. (8), respectively:

$$\boldsymbol{\mu} = \tilde{\boldsymbol{D}}^{-\frac{1}{2}} \boldsymbol{G} \tilde{\boldsymbol{D}}^{-\frac{1}{2}} \boldsymbol{H} \boldsymbol{W}_{\boldsymbol{\mu}}, \qquad (7)$$

$$\boldsymbol{\sigma} = \tilde{\boldsymbol{D}}^{-\frac{1}{2}} \boldsymbol{G} \tilde{\boldsymbol{D}}^{-\frac{1}{2}} \boldsymbol{H} \boldsymbol{W}_{\boldsymbol{\sigma}}, \qquad (8)$$

where $\boldsymbol{W}_{\boldsymbol{\mu}}$ and $\boldsymbol{W}_{\boldsymbol{\sigma}}$ are the weight matrices corresponding to the mean and variance, respectively. After inferring $\boldsymbol{Z}$, the decoder $p(\boldsymbol{G}|\boldsymbol{Z})$ is employed to reconstruct the original adjacency matrix $\boldsymbol{G}$ from $\boldsymbol{Z}$ through adopting the inner product by Eq. (9):

$$p(\boldsymbol{G}|\boldsymbol{Z}) = \prod_{i=1}^{n} \prod_{j=1}^{n} \zeta(\boldsymbol{z}_i^T \boldsymbol{z}_j), \qquad (9)$$

where $\zeta(\cdot)$ is a sigmoid function. $q(\boldsymbol{Z}|\boldsymbol{X}, \boldsymbol{G})$ and $p(\boldsymbol{G}|\boldsymbol{Z})$ are trained simultaneously by maximizing the evidence lower bound (ELBO), which can be formulated as Eq. (10):

$$\mathcal{L} = \mathbb{E}_{q(\boldsymbol{Z}|\boldsymbol{X}, \boldsymbol{G})}[\log p(\boldsymbol{G}|\boldsymbol{Z})] - KL(q(\boldsymbol{Z}|\boldsymbol{X}, \boldsymbol{G})||p(\boldsymbol{Z})), \qquad (10)$$

where $\mathbb{E}_{q(\boldsymbol{Z}|\boldsymbol{X}, \boldsymbol{G})}[\log p(\boldsymbol{G}|\boldsymbol{Z})]$ is the expectation of the binary cross entropy loss between the original adjacency matrix and the reconstructed matrix, and $KL(q(\boldsymbol{Z}|\boldsymbol{X}, \boldsymbol{G})||p(\boldsymbol{Z}))$ is the Kullback-Leibler divergence between $q(\boldsymbol{Z}|\boldsymbol{X}, \boldsymbol{G})$ and variational prior $p(\boldsymbol{Z})$.

To make new attributes align with the attribute conditional independence assumption and the Gaussian distribution required by GNB, we set the variational prior $p(\boldsymbol{Z})$ as an independent zero-mean Gaussian distribution with unit variances. VGAE ensures that the distribution of the embedding matrix closely approximates $p(\boldsymbol{Z})$. Each variate in $p(\boldsymbol{Z})$ corresponds to a new attribute in $\boldsymbol{Z}$, and thus the new attribute closely approximates the Gaussian distribution. The independent zero-mean in $p(\boldsymbol{Z})$ ensures that the covariance matrix of $p(\boldsymbol{Z})$ is a diagonal matrix, implying that covariances between new attributes approximate zero, i.e., new attributes are approximately independent of each other.

Based on $\mathcal{L}$, we update the parameters $\mathcal{W}$ of VGAE over $P$ iterations by Eq. (11):

$$\mathcal{W}_{p+1} = \mathcal{W}_p - \eta \frac{\partial \mathcal{L}}{\partial \mathcal{W}}, \qquad (11)$$

where $\mathcal{W}_p$ represents the parameters in the $p$-th iteration and $\eta$ represents the learning rate. When $p = 1$, $\mathcal{W}_p$ represents randomly initialized parameters.

Finally, after generating $\boldsymbol{Z}$, to enhance the identification abilities of original attributes, we augment original attributes by concatenating $\boldsymbol{Z}$ with $\boldsymbol{X}$. In this paper, we construct a VGAE consisting of three layers of neurons, with the number of neurons in each layer being $m$, $2m$ and $m$, respectively. Therefore, a total of $m$ new attributes are added. For each instance, we represent the $j$-th value of $\boldsymbol{z}_i$ as $z_{ij}$ and define $z_{ij}$ as the $(m + j)$-th attribute value of $\boldsymbol{x}_i$.

### 3.3. Attribute weighting

Although attribute augmentation enhances the identification abilities of original attributes, it may lead to redundant attributes. To alleviate the attribute redundancy, we can assign different weights for different augmented attributes. Then, we build attribute weighted GNB based on the weighted attributes and use Eq. (12) to predict the class label:

$$\hat{c}(\boldsymbol{x}) = \arg\max_{c \in C} \pi_c \prod_{j=1}^{2m} \theta_{a_j|c}^{w_j}, \qquad (12)$$

where $w_j$ is the weight of $A_j$. $\pi_c$ and $\theta_{a_j|c}$ are estimated by Eq. (13) and Eq. (14), respectively:

$$\pi_c = \frac{\sum_{i=1}^{n_t} I(c_i, c) + 1}{n_t + k}, \qquad (13)$$

$$\theta_{a_j|c} = \frac{1}{\sqrt{2\pi}\sigma_{cj}} \exp\left(-\frac{(a_j - \mu_{cj})^2}{2\sigma_{cj}^2}\right), \qquad (14)$$

where $k$ is the number of classes, $n_t$ is the number of training instances, $c_i$ is the class label of the $i$-th instance, $\mu_{cj}$ and $\sigma_{cj}$ are the mean and standard deviation of $A_j$ given $c$, respectively, and $I(\cdot)$ is a binary function, which takes the value 1 if its two parameters are identical and 0 otherwise.

Now, the only left question is how to calculate attribute weights. At first, we initialize each weight in the weight vector $\boldsymbol{w}$ using a random value between 0 and 1. Then, we optimize the initialized weights by the gradient descent search. The objective function of optimization is defined to maximize the CLL of the attribute weighted GNB, which can be formulated as Eq. (15):

$$\text{CLL}(\boldsymbol{w}) = \log \hat{P}(C|\boldsymbol{\mathcal{D}}_t, \boldsymbol{w}) = \sum_{i=1}^{n_t} \log \hat{P}(c_i|\boldsymbol{x}_i, \boldsymbol{w}), \qquad (15)$$

where $\boldsymbol{\mathcal{D}}_t$ is the training dataset, $\hat{P}(c_i|\boldsymbol{x}_i, \boldsymbol{w})$ is the posterior probability of $c_i$ estimated by the weighted GNB given $\boldsymbol{x}_i$ and $\boldsymbol{w}$, which can be formulated as Eq. (16):

$$\hat{P}(c_i|\boldsymbol{x}_i, \boldsymbol{w}) = \frac{\gamma_{c_i\boldsymbol{x}_i}(\boldsymbol{w})}{\sum_{c=1}^{k} \gamma_{c\boldsymbol{x}_i}(\boldsymbol{w})}, \qquad (16)$$

where $\gamma_{c\boldsymbol{x}_i}(\boldsymbol{w})$ is the product of $\pi_c$ and each $\theta_{a_j|c}^{w_j}$ of $\boldsymbol{x}_i$, which can be formulated as Eq. (17):

$$\gamma_{c\boldsymbol{x}_i}(\boldsymbol{w}) = \pi_c \prod_{j=1}^{2m} \theta_{a_j|c}^{w_j}. \qquad (17)$$

Before calculating the gradient of CLL($\boldsymbol{w}$) with respect to $w_j$, we can first calculate the gradient of $\gamma_{c\boldsymbol{x}_i}(\boldsymbol{w})$ with respect to $w_j$ as Eq. (18):

$$\begin{aligned}
\frac{\partial}{\partial w_j}\gamma_{c\boldsymbol{x}_i}(\boldsymbol{w}) &= \left(\pi_c \prod_{j'=1 \wedge j' \neq j}^{2m} \theta_{a_{j'}|c}^{w_{j'}}\right) \frac{\partial}{\partial w_j}\theta_{a_j|c}^{w_j} \\
&= \left(\pi_c \prod_{j'=1 \wedge j' \neq j}^{2m} \theta_{a_{j'}|c}^{w_{j'}}\right) \theta_{a_j|c}^{w_j} \log(\theta_{a_j|c}) \\
&= \gamma_{c\boldsymbol{x}_i}(\boldsymbol{w}) \log(\theta_{a_j|c}).
\end{aligned} \qquad (18)$$

Then, the gradient of CLL($\boldsymbol{w}$) with respect to $w_j$ can be represented as Eq. (19):

$$\begin{aligned}
&\frac{\partial}{\partial w_j}\text{CLL}(\boldsymbol{w}) \\
&= \frac{\partial}{\partial w_j} \sum_{i=1}^{n_t} \left(\log\left(\gamma_{c_i\boldsymbol{x}_i}(\boldsymbol{w})\right) - \log\left(\sum_{c=1}^{k} \gamma_{c\boldsymbol{x}_i}(\boldsymbol{w})\right)\right) \\
&= \sum_{i=1}^{n_t} \left(\frac{\gamma_{c_i\boldsymbol{x}_i}(\boldsymbol{w}) \log(\theta_{a_j|c_i})}{\gamma_{c_i\boldsymbol{x}_i}(\boldsymbol{w})} - \frac{\sum_{c=1}^{k} \gamma_{c\boldsymbol{x}_i}(\boldsymbol{w}) \log(\theta_{a_j|c})}{\sum_{c=1}^{k} \gamma_{c\boldsymbol{x}_i}(\boldsymbol{w})}\right) \\
&= \sum_{i=1}^{n_t} \left(\log(\theta_{a_j|c_i}) - \sum_{c=1}^{k} \hat{P}(c|\boldsymbol{x}_i, \boldsymbol{w}) \log(\theta_{a_j|c})\right).
\end{aligned} \qquad (19)$$

ICGNB can be partitioned into training (ICGNB-training) and classification (ICGNB-classification) algorithms. They are depicted by **Algorithms 2** and **3** provided in **Appendix A**. The time complexity is provided in **Appendix B**.

## 4. Experiments and results

We design two groups of experiments on 24 real-world datasets and a synthetic dataset, respectively. On the real-world datasets, we observe the classification performance of ICGNB compared to its five competitors and conduct an ablation study. On the synthetic dataset, we validate the effectiveness, independence and Gaussianity of the generated new attributes as well as the sensitivity of ICGNB.

### 4.1. Experiments on real-world datasets

From the real-world datasets published by the KEEL[1] dataset repository, we choose the whole 24 datasets only

---

[1]https://sci2s.ugr.es/keel/category.php?cat=clas

*Table 1.* Classification accuracy (%) comparisons for ICGNB versus WANBIA, CFWNB, AG-NBC, AE-NBC and GNB.

| Dataset | ICGNB | WANBIA | CFWNB | AG-NBC | AE-NBC | GNB |
|---|---|---|---|---|---|---|
| appendicitis | 89.09±7.66 | 88.18±7.66 | **90.00±6.03** | 83.64±6.80 | 83.18±10.77 | 86.36±6.10 |
| balance | 88.00±3.12 | 90.24±2.84 | 88.32±3.87 | **91.68±1.53** | 79.84±5.22 | 90.24±2.84 |
| banana | 69.53±0.80 | 62.00±0.91 | 59.74±1.48 | **84.80±3.83** | 62.41±3.19 | 62.00±0.91 |
| cleveland | 57.67±4.84 | **58.00±4.27** | 54.83±4.86 | 53.67±3.86 | 55.50±7.82 | 51.67±10.22 |
| ecoli | **79.12±5.46** | 79.12±6.06 | 60.29±11.60 | 70.00±6.35 | 78.53±2.81 | 60.74±6.61 |
| glass | **63.02±8.85** | 59.07±7.44 | 51.40±11.74 | 60.70±6.61 | 60.47±8.06 | 47.21±9.70 |
| iris | **96.33±3.14** | 96.00±3.27 | 95.33±3.71 | 90.33±6.90 | 91.00±7.31 | 95.33±3.71 |
| led7digit-01 | **72.40±4.27** | 70.40±5.90 | 64.20±9.11 | 71.30±4.73 | 71.70±3.26 | 63.30±12.12 |
| magic | **78.38±0.78** | 77.05±0.67 | 74.56±0.56 | 75.04±1.33 | 77.69±1.35 | 72.56±0.64 |
| movement_libras | 56.11±4.31 | 62.92±4.97 | 62.22±4.76 | 69.17±6.77 | **70.97±6.38** | 61.94±5.63 |
| phoneme | 76.46±1.10 | 75.91±1.40 | 76.85±1.58 | **77.22±1.69** | 76.91±1.62 | 75.97±1.65 |
| pima | 75.32±2.29 | **75.52±2.69** | 75.06±2.99 | 73.12±2.23 | 73.70±2.94 | 74.61±3.45 |
| ring | **97.98±0.36** | 97.90±0.20 | 97.96±0.30 | 93.34±1.18 | 94.73±0.51 | 97.92±0.28 |
| segment | **90.41±1.59** | 88.81±1.26 | 80.52±1.26 | 88.01±1.76 | 83.35±2.60 | 79.42±1.48 |
| sonar | **78.57±5.73** | 78.33±5.05 | 67.86±5.46 | 76.67±7.81 | 67.15±8.05 | 66.67±5.11 |
| spambase | **90.67±1.08** | 89.99±1.08 | 83.71±1.26 | 86.52±1.83 | 86.51±0.98 | 82.08±1.25 |
| texture | **96.84±0.52** | 84.47±1.00 | 78.35±1.38 | 94.91±0.78 | 94.22±0.74 | 77.45±1.39 |
| titanic | 77.41±1.19 | **77.64±1.21** | 76.98±0.89 | 75.51±1.70 | 76.98±0.89 | 76.98±0.89 |
| twonorm | 97.66±0.23 | **97.72±0.27** | 97.71±0.29 | 96.37±0.61 | 95.31±0.91 | 97.70±0.28 |
| wdbc | **96.32±1.51** | 96.40±1.54 | 93.95±1.90 | 94.65±1.69 | 85.53±2.97 | 92.98±2.29 |
| wine | 96.94±1.94 | 97.50±1.50 | 96.94±1.94 | **97.50±1.94** | 92.22±5.53 | 97.50±1.50 |
| winequality-red | **59.53±2.56** | 58.44±1.78 | 58.47±1.49 | 58.84±3.19 | 57.16±2.93 | 54.72±2.56 |
| winequality-white | **52.64±1.27** | 52.21±1.51 | 49.23±1.14 | 51.02±1.94 | 51.51±1.43 | 44.38±1.61 |
| yeast | **56.53±3.26** | 54.28±3.20 | 18.22±3.83 | 50.03±3.07 | 55.49±1.87 | 14.41±3.38 |
| (**W / T / L**) | | 17/0/7 | 18/1/5 | 19/0/5 | 22/0/2 | 20/0/4 |
| **Average** | 78.87 | 77.84 | 73.03 | 77.67 | 75.92 | 71.84 |

*Table 2.* Wilcoxon tests for ICGNB versus WANBIA, CFWNB, AG-NBC, AE-NBC and GNB.

| Algorithm | ICGNB | WANBIA | CFWNB | AG-NBC | AE-NBC | GNB |
|---|---|---|---|---|---|---|
| ICGNB | - | ○ | ○ | ○ | ○ | ○ |
| WANBIA | ● | - | ○ | | ○ | ○ |
| CFWNB | ● | ● | - | ● | | ○ |
| AG-NBC | ● | | ○ | - | | ○ |
| AE-NBC | ● | ● | | | - | ○ |
| GNB | ● | ● | ● | ● | | - |

containing numerical attributes, which represent a wide range of domains and data characteristics. The detailed description of these datasets is provided in **Appendix C**. In our experiments, we apply z-score normalization (Patro & Sahu, 2015) to attributes before inputting them into algorithms. We compare the classification accuracy (%) of ICGNB with its five competitors and ablation variants on these datasets by running 10 separate stratified hold-out validations. In each validation, we use stratified sampling to split the dataset into a training set (80%) and a testing set (20%).

**Classification performance.** We compare ICGNB with its five competitors, including weighting attributes to alleviate naive Bayes' independence assumption (WANBIA) (Zaidi et al., 2013), correlation-based feature weighting filter for naive Bayes (CFWNB) (Jiang et al., 2019a), attribute grouping-based naive Bayesian classifier (AG-NBC) (He et al., 2023), auto-encoding naive Bayesian classifier (AE-

NBC) (Ou et al., 2022) and GNB. WANBIA and CFWNB are two existing state-of-the-art improved algorithms of NB focusing on nominal attributes with attribute weighting. In our experiments, to enable them to handle numeric attributes directly, we replace the NB in them with GNB. AG-NBC and AE-NBC are two existing state-of-the-art improved algorithms of NB focusing on numerical attributes with attribute generation. Different from our proposed ICGNB, they generate independent attribute groups or attributes to align with the attribute conditional independence assumption. Among all competitors, GNB is the baseline. We implement ICGNB, WANBIA, CFWNB, AG-NBC, AE-NBC and GNB by using Python, respectively. In ICGNB, the number of iterations $P$ is 500, the learning rate $\eta$ is 0.01, and $\sigma_{1i} \approx 1000\sigma_{2i}$. In AE-NBC, the number of iterations and the learning rate are the same as those of ICGNB. In AG-NBC, the stopping threshold $\delta$ is 0.001 and the regular-

ization factors $\varepsilon_1$, $\varepsilon_2$ are 1, -1, respectively.

Table 1 shows the detailed classification accuracy of each algorithm on each dataset. We bold the highest classification accuracy on each dataset to show the algorithm which performs best in that dataset. If two algorithms have the same accuracy on the same dataset, we bold the accuracy with a small variance. The Win / Tie / Lose values and the averages are summarized at the bottom of the table. The Win / Tie / Lose (**W / T / L**) values of each entry in the table provide a concise summary of how ICGNB fares against its competitors. They imply that, compared to its competitors, ICGNB wins on **W** datasets, ties on **T** datasets, and loses on **L** datasets. The average (arithmetic mean) of each algorithm across all datasets provides a gross indicator of the relative performance. Based on the classification accuracy results presented in Table 1, we employ the Wilcoxon signed-ranks test (Demsar, 2006) to conduct a comprehensive comparison of each pair of algorithms. This non-parametric statistical test ranks the differences in performance for each dataset, ignoring the signs, and compares the ranks for positive and negative differences. The comparison results are summarized in Table 2, where ● indicates that the algorithm in the column outperforms the one in the corresponding row, and ○ signifies the opposite. The lower-diagonal significance level is $\alpha = 0.05$, while the upper-diagonal level is $\alpha = 0.1$. These results show that our proposed ICGNB significantly outperforms all the other competitors. Now, we summarize the highlights as follows:

(1) Compared to WANBIA and CFWNB, ICGNB wins on 17 and 18 datasets and only loses on 7 and 5 datasets, respectively. ICGNB obtains much better performance than the existing state-of-the-art competitors with attribute weighting. This indicates the effectiveness of using new attributes containing the correlations among instances.

(2) Compared to AG-NBC and AE-NBC, ICGNB wins on 19 and 22 datasets and loses on 5 and 2 datasets, respectively. ICGNB performs much better than existing state-of-the-art competitors with attribute generation. This indicates the effectiveness of alleviating the attribute redundancy by assigning different weights for different augmented attributes.

(3) Compared to GNB, ICGNB wins on 20 datasets and loses only on 4 datasets. This verifies that ICGNB demonstrates significant effectiveness in enhancing the performance of NB and our proposed ICG-based representation learning method is powerful.

(4) The average classification accuracy of ICGNB on 24 datasets is 78.87%, which is remarkably higher than those of WANBIA (77.84%), CFWNB (73.03%), AG-NBC (77.67%), AE-NBC (75.92%) and GNB (71.84%), respectively. This proves that ICGNB is generally the best among all the competitors.

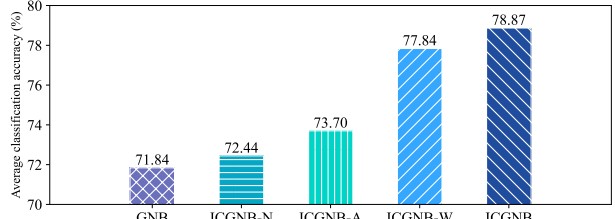

*Figure 2.* Results of the ablation study.

(5) According to the Wilcoxon signed-ranks test results presented in Table 2, ICGNB significantly outperforms all existing state-of-the-art competitors whether $\alpha = 0.05$ or $\alpha = 0.1$, which strongly validates the classification performance of ICGNB.

**Ablation study.** To thoroughly analyze whether each part in ICGNB takes effect, we conduct an ablation study about three ablation variants. They retain different parts in ICGNB to observe the effectiveness of specific parts. ICGNB-N is the first variant, in which the part of attribute generation is the same as that of ICGNB while attribute augmentation and weighting are removed. ICGNB-A is the second variant, in which the parts of attribute generation and augmentation are the same as those of ICGNB while attribute weighting is removed. In addition to ICGNB-N and ICGNB-A, WANBIA can serve as another ablation variant of ICGNB. We represent it as ICGNB-W, in which the part of attribute weighting is the same as that of ICGNB while attribute generation and augmentation are removed. According to the results shown in Figure 2, we summarize the conclusions as follows: (1) The accuracy of ICGNB-A is higher than that of GNB and ICGNB-N, which indicates that attribute augmentation is necessary for leveraging original attributes and new attributes simultaneously. (2) The accuracy of ICGNB is higher than that of ICGNB-A, which proves that attribute weighting is necessary for alleviating the attribute redundancy. (3) The accuracy of ICGNB is also higher than that of ICGNB-W, which suggests that attribute generation and augmentation are necessary for enhancing the identification abilities of original attributes.

### 4.2. Experiments on synthetic dataset

To further verify the effectiveness, independence and Gaussianity of the generated new attributes as well as the sensitivity of ICGNB, we design another group of experiments on a synthetic dataset. The synthetic dataset contains 2 classes, 100 instances and 50 attributes. The synthesis process initially creates Gaussian clusters around the vertices of a 40-dimensional hypercube and assigns an equal number of clusters to each class. By sampling from these clusters, instances containing 40 informative attributes can be obtained. Then, 10 redundant attributes are generated by random lin-

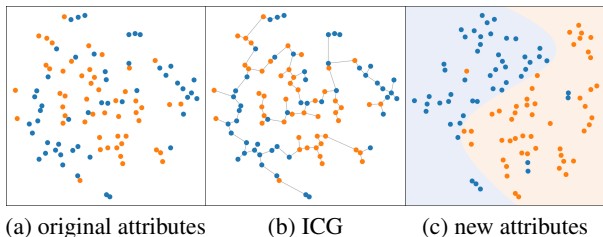

(a) original attributes    (b) ICG    (c) new attributes

*Figure 3.* Class distributions of the synthetic dataset.

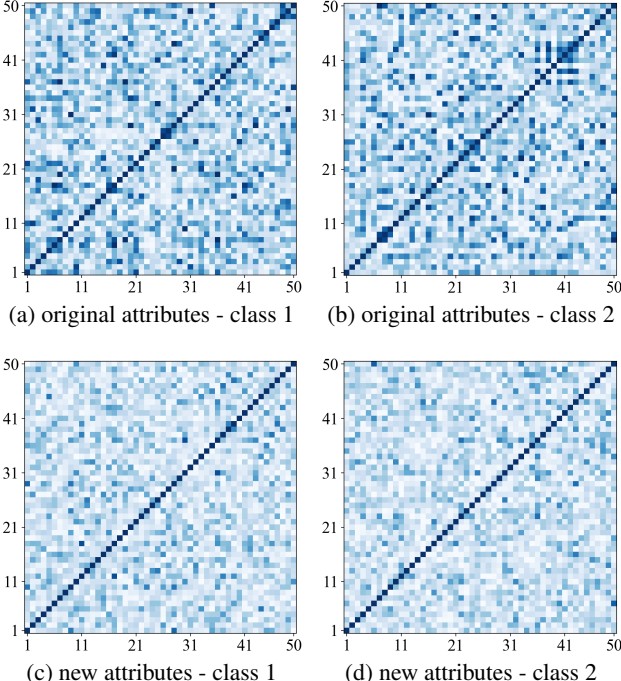

(a) original attributes - class 1    (b) original attributes - class 2

(c) new attributes - class 1    (d) new attributes - class 2

*Figure 4.* Pearson coefficients between original attributes and new attributes in two classes.

ear combinations of these informative attributes. To shift the distribution of each attribute away from the Gaussian distribution, attribute values are mapped by using the transform function as Eq. (20):

$$a'_s = sign(a_s) \cdot a_s{}^2, \tag{20}$$

where $a'_s$ is the mapped attribute value, $a_s$ is an original attribute value, $sign(\cdot)$ is a signed function taking 0, 1, -1 if $a_s$ is zero, positive, negative, respectively.

**Effectiveness.** To facilitate clear observations of the class distributions of the synthetic dataset, we employ the t-SNE algorithm (van der Maaten & Hinton, 2008) to map the synthetic dataset to a two-dimensional dataset. Figure 3 shows the class distributions with original attributes, ICG and augmented attributes. For ICG, we hide the self-connecting edges of instances. In Figure 3b, we can see that ICG effec-

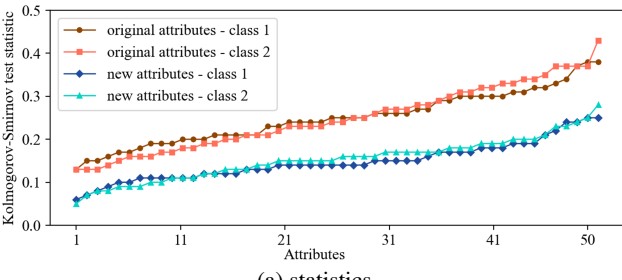

(a) statistics

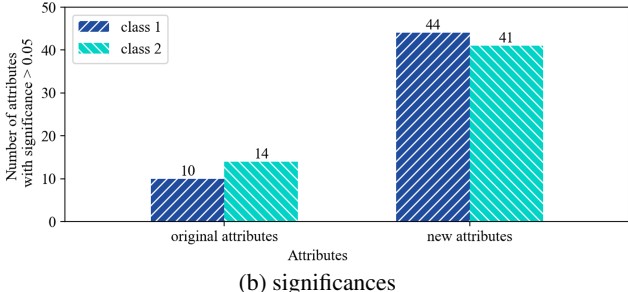

(b) significances

*Figure 5.* Gaussianity of attribute comparisons of the Kolmogorov-Smirnov tests.

tively connects instances of the same class. This is helpful in representing the correlations among instances on original attributes. In Figure 3c, instances of the same classes are generally distributed within the same regions, and instances of different classes can be separated by a clear boundary. The class distribution with augmented attributes demonstrates significant distinguishability compared to that with original attributes in Figure 3a, in which instances of different classes are scattered. This suggests that augmented attributes containing new attributes are more effective for classification than original attributes.

**Independence.** To investigate whether new attributes align better with the attribute conditional independence assumption than original attributes, we calculate the absolute value of the Pearson coefficient (Cohen & Israel, 2009) as the correlation between each pair of attributes given $c$. We plot the correlations as heat maps, in which the lower correlations correspond to the lighter colors, as shown in Figure 4. The elements on the diagonal indicate the correlations of the attributes to themselves, which are excluded from our analysis and are set to the darkest color. Figure 4c and Figure 4d are much lighter in color than Figure 4a and Figure 4b, indicating that new attributes have lower correlations with each other given $c$ than original attributes. Therefore, new attributes align better with the attribute conditional independence assumption than original attributes.

**Gaussianity.** We use the Kolmogorov-Smirnov test to investigate whether new attributes conform to the Gaussian distribution required by GNB better than original attributes. The Kolmogorov-Smirnov test can compare whether there is

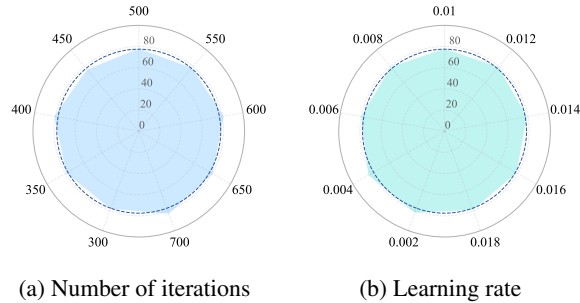

(a) Number of iterations      (b) Learning rate

*Figure 6.* Average classification accuracy (%) of ICGNB with different parameters.

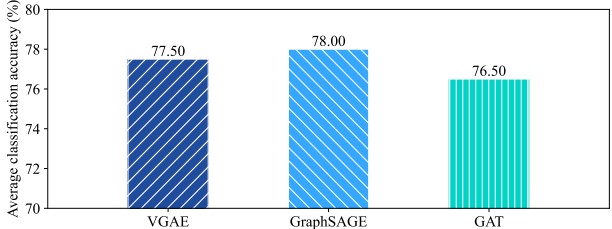

*Figure 7.* Average classification accuracy (%) of ICGNB with different graph convolution functions.

a significant difference between two distributions. On original attributes and new attributes, we calculate the statistics and significances between the conditional probability distribution of each attribute given $c$ and the Gaussian distribution which has the same mean and variance as the distribution to be tested. The higher statistics indicate the greater difference between the two distributions. A significance level below 0.05 indicates that the conditional probability distribution is significantly different from the Gaussian distribution. Conversely, the conditional probability distribution demonstrates significant Gaussianity. We present the comparisons of sorted statistics for original attributes and new attributes in two classes in Figure 5a. The comparisons show that new attributes consistently exhibit lower statistics compared to original attributes in both class 1 and class 2. This indicates that new attributes are generally closer to the Gaussian distribution than original attributes. We present the number of attributes with significance levels exceeding 0.05 in two classes in Figure 5b. It can be found that there are 10 and 14 original attributes in class 1 and class 2 demonstrating significant Gaussianity, respectively. In contrast, there are 44 and 41 new attributes in class 1 and class 2 demonstrating significant Gaussianity, respectively. These results indicate that new attributes conform to the Gaussian distribution required by GNB better than original attributes.

**Sensitivity.** To analyze the sensitivity of ICGNB with different parameters and graph convolution functions, we observe its classification performance on the synthetic dataset, and the experimental setup remains identical to that outlined in subsection 4.1. In ICGNB, the main adjustable parameters are the number of iterations $P$ and the learning rate $\eta$. We conduct two parameter sensitivity analysis experiments by fixing one parameter while changing another one, and then observe the classification performance. The detailed experimental results are shown in Figures 6a and 6b, where the blue circle corresponds to $P = 500$ and $\eta = 0.01$. From these results, we can see that the average classification accuracy is consistently near the blue circle, displaying no significant fluctuation with the changing of $P$ or $\eta$. This

demonstrates that ICGNB is not sensitive to $P$ and $\eta$. In addition to changing the adjustable parameters, we replace the graph convolution function of VGAE in ICGNB with those of GraphSAGE (Hamilton et al., 2017) and Graph Attention Network (GAT) (Velickovic et al., 2018), and then observe the classification performance. The detailed experimental results are shown in Figure 7. From Figure 7, we can see that the average classification accuracy corresponding to three graph convolution functions are 77.50%, 78.00% and 76.50%, respectively. These results show that using the graph convolution function of GraphSAGE slightly increases the performance of ICGNB, while using that of GAT slightly reduces it. This demonstrates that ICGNB is not sensitive to the graph convolution function.

## 5. Conclusion and future work

To leverage the correlations among instances and then enhance the identification abilities of original attributes, we develop an instance correlation graph (ICG)-based representation learning method. Based on this method, we propose a novel algorithm called instance correlation graph-based naive Bayes (ICGNB). In ICGNB, we first construct an ICG by using original attributes to represent the correlations among instances. Then, we employ a variational graph auto-encoder (VGAE) to generate new attributes from the ICG and then augment original attributes using the generated new attributes. Finally, we weight each augmented attribute to alleviate attribute redundancy and then build GNB on the weighted attributes.

The strategy of constructing ICG is critical, which determines the precision and effectiveness of representing the correlations among instances. In this paper, we do not incorporate class labels when constructing ICG, leading to the omission of the supervised information. Exploring how to construct ICG with supervised information is the main direction for future work. In addition, we construct ICG using a greedy search strategy, which increases the computational cost of ICGNB. In this context, exploring how to design a strategy with lower computational cost is another direction for future work.

## Acknowledgments

The work was partially supported by National Natural Science Foundation of China (62276241) and Hubei Provincial Collaborative Innovation Center for Basic Education Information Technology Services (OFHUE202312).

## Impact Statement

This paper presents work whose goal is to advance the field of Machine Learning. There are many potential societal consequences of our work, none which we feel must be specifically highlighted here.

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

# A. Training (ICGNB-training) and classification (ICGNB-classification) algorithms of ICGNB.

---

**Algorithm 2** ICGNB-training($\mathcal{D}$)

---

1: **Input:** $\mathcal{D} = \{\boldsymbol{X}, \boldsymbol{c}\}$ - the dataset.
2: **Output:** $\boldsymbol{Z}$ - the new attribute matrix, $\boldsymbol{w}$ - the weight vector.
3: Construct ICG by **Algorithm 1**;
4: Transform ICG into an adjacency matrix $\boldsymbol{G}$;
5: Initialize the parameters of VGAE as $\mathcal{W}_1$;
6: **for** $p = 1$ to $P$ **do**
7:     Generate the embedding matrix by Eq. (4) with $\mathcal{W}_p$;
8:     Reconstruct the original adjacency matrix by Eq. (9);
9:     Calculate the loss by Eq. (10);
10:     Update the parameters by Eq. (11);
11: **end for**
12: Generate the new attribute matrix $\boldsymbol{Z}$ by Eq. (4) with $\mathcal{W}_{P+1}$;
13: **for** $i = 1$ to $n_t$ **do**
14:     **for** $j = 1$ to $m$ **do**
15:         Define the $z_{ij}$ as the $(m + j)$-th attribute value of $\boldsymbol{x}_i$;
16:     **end for**
17: **end for**
18: **for** $c = 1$ to $k$ **do**
19:     Estimate the prior probability $\pi_c$ by Eq. (13);
20:     **for** $j = 1$ to $2m$ **do**
21:         Estimate the conditional probability $\theta_{a_j|c}$ by Eq. (14);
22:     **end for**
23: **end for**
24: Initialize each weight in the weight vector $\boldsymbol{w}$;
25: Optimize the initialized weight vector $\boldsymbol{w}$ by Eqs. (15) - (19);
26: **return** $\boldsymbol{Z}, \boldsymbol{w}$.

---

**Algorithm 3** ICGNB-classification($\boldsymbol{Z}, \boldsymbol{w}, \boldsymbol{x}$)

---

1: **Input:** $\boldsymbol{Z}$ - the new attribute matrix, $\boldsymbol{w}$ - the weight vector, $\boldsymbol{x}$ - a test instance.
2: **Output:** $\hat{c}(\boldsymbol{x})$ - the predicted class label of $\boldsymbol{x}$.
3: Extract the embedding vector $\boldsymbol{z}$ corresponding to $\boldsymbol{x}$ from $\boldsymbol{Z}$;
4: **for** $j = 1$ to $m$ **do**
5:     Define $z_{ij}$ as the $(m + j)$-th attribute value of $\boldsymbol{x}$;
6: **end for**
7: **for** $c = 1$ to $k$ **do**
8:     Estimate the prior probability $\pi_c$ by Eq. (13);
9:     **for** $j = 1$ to $2m$ **do**
10:         Estimate the conditional probability $\theta_{a_j|c}$ by Eq. (14);
11:     **end for**
12: **end for**
13: Predict the class label $\hat{c}(\boldsymbol{x})$ of $\boldsymbol{x}$ by Eq. (12);
14: **return** $\hat{c}(\boldsymbol{x})$.

---

## B. Time complexity of ICGNB.

In **Algorithm 1**, line 3 constructs a full connection graph with a time complexity of $O(n^2)$. Lines 4-8 calculate the Euclidean distance with a time complexity of $O(n^2m)$. Line 9 sorts edges with a time complexity of $O(n^2logn)$. Lines 10-18 add edges to ICG with a time complexity of $O(n^2\alpha(n))$, where $O(\alpha(n))$ is the time complexity of checking if two vertices are reachable. Lines 19-21 add self-connecting edges with a time complexity of $O(n)$. Due to $m$ usually being greater than $logn$ and $\alpha(n)$, considering only the highest-order terms, the overall time complexity of **Algorithm 1** is $O(n^2m)$.

In **Algorithm 2**, line 3 constructs ICG with a time complexity of $O(n^2m)$. Lines 4-5 transform ICG into $G$ and initialize the parameters in VGAE with a time complexity of $O(n^2)$. Lines 6-11 train a VGAE with a time complexity of $O(P(nm^2 + n^2m))$. Line 12 generates new attributes with a time complexity of $O(nm^2)$. Lines 13-17 augment original attributes with a time complexity of $O(nm)$. Lines 18-23 train a GNB with a time complexity of $O(knm)$. Lines 24-25 weight augmented attributes with a time complexity of $O(\beta(m))$, where $\beta(m)$ has a linear relationship with $m$. Due to $n$ usually being greater than $m$, considering only the highest-order terms, the overall time complexity of **Algorithm 2** is $O(Pn^2m)$.

In **Algorithm 3**, line 3 extracts the embedding vector with a time complexity of $O(1)$. Lines 4-6 augment original attributes with a time complexity of $O(m)$. Lines 7-12 estimate $\pi_c$ and each $\theta_{a_j|c}$ with a time complexity of $O(km)$. Line 13 predicts $\hat{c}(x)$ with a time complexity of $O(m)$. Considering only the highest-order terms, the overall time complexity of **Algorithm 3** is $O(km)$.

## C. Descriptions of 24 real-world datasets used in experiments.

*Table 3.* Descriptions of 24 real-world datasets used in experiments. "#Attributes" denotes the number of attributes, "#Classes" denotes the number of classes and "#Instances" denotes the number of instances.

| Dataset | #Attributes | #Classes | #Instances |
|---|---|---|---|
| appendicitis | 7 | 2 | 106 |
| balance | 4 | 3 | 625 |
| banana | 2 | 2 | 5300 |
| cleveland | 13 | 5 | 297 |
| ecoli | 7 | 8 | 336 |
| glass | 9 | 7 | 214 |
| iris | 4 | 3 | 150 |
| led7digit-01 | 7 | 10 | 500 |
| magic | 10 | 2 | 19020 |
| movement_libras | 90 | 15 | 360 |
| phoneme | 5 | 2 | 5404 |
| pima | 8 | 2 | 768 |
| ring | 20 | 2 | 7400 |
| segement | 19 | 7 | 2310 |
| sonar | 60 | 2 | 208 |
| spambase | 57 | 2 | 4597 |
| texture | 40 | 11 | 5500 |
| titanic | 3 | 2 | 2201 |
| twonorm | 20 | 2 | 7400 |
| wdbc | 30 | 2 | 569 |
| wine | 13 | 3 | 178 |
| winequality-red | 11 | 11 | 1599 |
| winequality-write | 11 | 11 | 4898 |
| yeast | 8 | 10 | 1484 |

