# OpenReview forum: "Instance Correlation Graph-based Naive Bayes"
_ICML.cc/2025/Conference — ICML 2025 spotlightposter_

### Official Review · Reviewer_XBWa · 2025-03-09

**Overall Recommendation:** 4

**Summary:**

The authors propose a novel algorithm called instance correlation graph-based naive Bayes (ICGNB), which can work with numerical attributes and utilize the correlations among instances. The average classification accuracy of ICGNB on 24 datasets is higher than the best competitors.

**Claims And Evidence:**

The results on synthetic data demonstrate that the newly generated attributes are more effective than original ones. The authors consider all feasible datasets from KEEL for evaluation, and thus I think they did not hide negative results. Hence, I'm overall satisfied with the improvements, despite that the proposed ICGNB does not always perform the best.

**Essential References Not Discussed:**

No, to my knowledge.

**Experimental Designs Or Analyses:**

The experimental designs of both real and synthetic datasets are checked. I think the ablation study part can be extended to include more variants.

**Methods And Evaluation Criteria:**

The main framework of ICGNB in Figure 1 is intuitive. The optimization of ELBO aligns with the choice of VGAE. The choice of datasets is ok to me, since ALL datasets containing only numerical attributes from KEEL are considered.

**Other Comments Or Suggestions:**

I would suggest to put the columns of ICGNB and ICGNB-A together for easy comparison.

I wonder whether the formulas in the introduction section are necessary, since they are not used in the motivation illustration of ICGNB.

**Other Strengths And Weaknesses:**

No other strengths and weaknesses need to be highlighted.

**Questions For Authors:**

1.How about the performance if other graph convolution functions are used?

**Relation To Broader Scientific Literature:**

NA

**Theoretical Claims:**

NA

---

> ### Author Rebuttal · Authors · 2025-04-01
>
> **Questions For Authors:** How about the performance if other graph convolution functions are used?
>
> **Author Response:** Thanks for your valuable comments. In addition to the graph convolution function used in VGAE, some other graph convolution functions in GraphSAGE and GAT can also be used. To address the reviewer’s concerns, we perform experiments by replacing the graph convolution function of VGAE with those of GraphSAGE and GAT to evaluate the performance of ICGNB. We perform a stratified hold-out validation on 24 real-world datasets. All other experimental settings remain consistent with those described in the paper. The average classification accuracy of ICGNB with different graph convolution functions across the 24 datasets is summarized as follows:
> ||ICGNB_VGAE|ICGNB_GraphSAGE|ICGNB_GAT|
> |:--:|:--:|:--:|:--:|
> | Average Accuracy (%)|80.22|80.46|79.22|
> ||
>
> These results show that using the graph convolution function of GraphSAGE is also effective, but using that of GAT slightly reduces our ICGNB’s performance.
>
> **Experimental Designs Or Analyses:** The experimental designs of both real and synthetic datasets are checked. I think the ablation study part can be extended to include more variants.
>
> **Author Response:** Thanks for your valuable comments. We agree that including additional ablation variants can provide a more thorough analysis of the effectiveness of each part in ICGNB. To address the reviewer’s concerns, we have introduced two new ablation variants and conducted a more comprehensive ablation study using the experimental setup described in Section 4.1. The two new variants are denoted as ICGNB-G and ICGNB-GW. Here, ICGNB-G only retains the part of attribute generation and removes attribute augmentation and weighting. ICGNB-GW retains the part of attribute generation and weighting and removes original attributes. Meanwhile, the existing two ablation variants are denoted as ICGNB-A and ICGNB-W. ICGNB-A only retains the part of attribute generation and attribute augmentation and removes attribute weighting. ICGNB-W retains the part of attribute weighting and removes attribute generation and attribute augmentation. The experimental results are as follows:
> ||ICGNB|ICGNB\-A|ICGNB\-W|ICGNB\-G|ICGNB\-GW|
> |:--:|:--:|:--:|:--:|:--:|:--:|
> |Average Accuracy (%)|78.87|73.70|77.84|72.44|72.61|
> ||
>
> From these results, it can be found that ICGNB-A achieves higher accuracy than ICGNB-G, demonstrating that attribute augmentation is essential for ICGNB. ICGNB-GW achieves higher accuracy than ICGNB-G, confirming that attribute weighting is necessary for new attributes. Meanwhile, ICGNB consistently outperforms all its variants, further validating the rationality of ICGNB. In the final version of the paper, we will add these two new variants into our ablation study. Thanks again for your valuable comments.
>
> **Other Comments Or Suggestions:**
>
> I would suggest to put the columns of ICGNB and ICGNB-A together for easy comparison.
>
> I wonder whether the formulas in the introduction section are necessary since they are not used in the motivation illustration of ICGNB.
>
> **Author Response:**
> Thanks for your valuable comments. In the final version of the paper, we will put the columns of ICGNB and ICGNB-A together for easy comparison.
>
> In our paper, the introduction section includes two formulas, Eqs (1) and (2), which are used by Bayesian networks and NB to classify instances, respectively. As the starting point of our work, they demonstrate that NB relies on the attribute conditional independence assumption (ACIA). Considering that ACIA is difficult to hold in real-world scenarios, numerous improved algorithms have subsequently been proposed. However, these algorithms overlook leveraging the correlations among instances to improve NB, which motivates us to propose the ICGNB in this paper. Therefore, the formulas in the introduction section are necessary. Thanks again for your valuable comments.

---

### Official Review · Reviewer_DoNX · 2025-03-09

**Overall Recommendation:** 4

**Summary:**

This paper proposes a novel instance correlation graph (ICG) based Naïve Bayes classification framework. It first constructs an ICG from original attributes, and then employs a variational graph autoencoder (VGAE) to generate embeddings based on both ICG and original attributes. Extensive experiments have been conducted to verify the effectiveness of the proposed method.

**Claims And Evidence:**

Yes. The authors claim that they improve the performance of Naïve Bayes classifier by leveraging the correlations among instances, which is supported by clear and convincing theoretical analysis and experiments.

**Essential References Not Discussed:**

No. All essential references are currently cited/discussed in the paper.

**Experimental Designs Or Analyses:**

Yes. I have checked the soundness of the experimental settings and results analysis. Twenty-four real-world datasets were adopted for evaluating the classification performance of the proposed method, and synthetic data was used to verify the independence and Gaussianity of the attributes generated by ICG. These experimental designs and analyses look sound.

**Methods And Evaluation Criteria:**

Yes. The proposed method and evaluation criteria in the paper make sense for the problem and application.

**Other Comments Or Suggestions:**

I do not have any additional comments or suggestions for the authors.

**Other Strengths And Weaknesses:**

S1. Unlike most variants of Naïve Bayes that assume instance independence, this paper enhances Naïve Bayes based on the correlations among instances, which seems both reasonable and effective.

S2. The proposed model ICGNB contains an attribute weighting module to reduce the redundancy information from the correlations of instances, which is interesting and effective.

S3. Extensive experiments on several real-world and synthetic datasets have been conducted and the results verify the effectiveness, independence and Gaussianity of the proposed model.

S4. The organization of the paper is quite good and it is easy to follow.

W1. The ICG construction algorithm appears to employ a greedy strategy. Thus, the correctness of this algorithm should be discussed more clearly.

W2. The authors claim that using a VGAE to generate augmented attributes from ICG and reweighting these attributes is effective for improving classification accuracy. However, this technique is similar to Graph Attention Networks (GAT), so the advantages of the proposed method over GAT should be clearly explained.

**Questions For Authors:**

Q1. Constructing a full connection graph of instances based on Euclidean distance is time consuming and usually introduces noisy edges, how to address these issues?

Q2. The ICG construction algorithm adopts a greedy strategy, and its output is similar to a minimum spanning tree of the full connection graph. Moreover, the class labels of training instances are not incorporated into the ICG construction, which might lead to the omission of supervised information. Therefore, the correctness and advantages of this algorithm should be thoroughly discussed.

**Relation To Broader Scientific Literature:**

This paper proposes a novel Naïve Bayes (NB) classifier, named ICGNB, to improve the performance of NB on numerical attributes by effectively utilizing the correlations among instances with VGAE.

**Theoretical Claims:**

Yes. I have checked the correctness of theoretical claims in this paper.

---

> ### Author Rebuttal · Authors · 2025-04-01
>
> **Q1:** Constructing a full connection graph of instances based on Euclidean distance is time consuming and usually introduces noisy edges, how to address these issues?
>
> **Author Response to Q1:** Thanks for your valuable comments. In our paper, to construct the instance correlation graph (ICG), we first define a full connection graph containing $n(n-1)/2$ edges among all instances. Then, to reduce time costs, we only select $n-1$ edges from this full connection graph to construct ICG. As a result, the ICG we constructed is a sparse graph, with most noisy edges from the fully connected graph removed. In the final version of the paper, we will include the above explanations to clarify this point. Thanks again for your valuable comments.
>
> **Q2 and W1:** The ICG construction algorithm adopts a greedy strategy, and its output is similar to a minimum spanning tree of the full connection graph. Moreover, the class labels of training instances are not incorporated into the ICG construction, which might lead to the omission of supervised information. Therefore, the correctness and advantages of this algorithm should be thoroughly discussed.
>
> **Author Response to Q2 and W1:** Thanks for your valuable comments. In ICGNB, VGAE requires a graph that includes both training and test instances. Since the class labels of test instances are unavailable during the training stage, their supervised information cannot be incorporated into the construction of the ICG. To ensure consistency between training and test instances, the supervised information of training instances is also excluded from the ICG construction. Although supervised information is not used during ICG construction, it is incorporated into the attribute weighting stage of ICGNB. Therefore, the overall ICGNB framework fully considers supervised information. In the final version of the paper, we will include the above explanations to clarify this point. Thanks again for your valuable comments.
>
> **W2:** The authors claim that using a VGAE to generate augmented attributes from ICG and reweighting these attributes is effective for improving classification accuracy. However, this technique is similar to Graph Attention Networks (GAT), so the advantages of the proposed method over GAT should be clearly explained.
>
> **Author Response to W2:** Thanks for your valuable comments. The advantages of ICGNB over GAT are summarized in the following two aspects. First, GAT fails to retain the original attributes during training, thereby losing their interpretability. In contrast, ICGNB simultaneously utilizes both the original attributes and the newly generated attributes of all instances, which preserves the interpretability of the original attributes while capturing potential correlations among instances through the new attributes. Second, GAT applies a local attention mechanism to assign weights to the neighbors of each node. However, ICGNB applies a global weighting strategy to all 2$m$ augmented attributes, enabling it to capture the overall characteristics of the dataset more effectively. By combining these two aspects, ICGNB demonstrates clear advantages over GAT. We will include a discussion on these advantages in the final version of the paper. Thanks again for your valuable comments.

---

### Official Review · Reviewer_XTK4 · 2025-03-11

**Overall Recommendation:** 4

**Summary:**

This paper presents ICGNB, an enhanced Naïve Bayes method based on the Instance Correlation Graph (ICG). ICGNB leverages ICG to capture instance correlations, employs a Variational Graph Autoencoder (VGAE) to generate new attributes, and optimizes attribute weighting using Conditional Log-Likelihood (CLL). Experimental results demonstrate that ICGNB achieves an average classification accuracy of 78.87% across 24 real-world datasets, significantly outperforming existing methods. Its effectiveness is further validated through Wilcoxon statistical tests. Ablation studies and synthetic dataset analysis further confirm that ICGNB generates attributes with higher independence and Gaussianity, leading to improved classification performance.

**Claims And Evidence:**

Yes, the paper demonstrates through Wilcoxon statistical tests that incorporating instance correlations enhances Naïve Bayes' ability to handle numerical attributes. Additionally, Pearson correlation heatmaps reveal a significant reduction in correlation among the newly generated attributes, making them more aligned with the Naïve Bayes assumption.

**Essential References Not Discussed:**

The paper does not discuss classification methods based on GNN and VGAE, such as GraphSAGE [1], which have had a significant impact in the field of graph representation learning and are closely related to ICGNB’s graph-based representation approach. Additionally, the paper does not cite more advanced probabilistic graphical models, which could be relevant to optimizing the compatibility of ICGNB with the NB assumption. Including discussions on these methods could provide a broader contextual understanding and highlight potential improvements to the proposed approach.

- [1] Hamilton W, Ying Z, Leskovec J. Inductive representation learning on large graphs[J]. Advances in neural information processing systems, 2017, 30.

**Experimental Designs Or Analyses:**

At present, the experiments are all reasonable, but there is a lack of parametric sensitivity experiments.

**Methods And Evaluation Criteria:**

This approach is meaningful as it utilizes the ICG to capture instance relationships, overcoming the limitation of NB, which relies solely on independent attributes. By leveraging VGAE to generate new attributes, the method enhances the representational capacity of the original attributes while ensuring their independence.

**Other Comments Or Suggestions:**

None.

**Other Strengths And Weaknesses:**

Strengths:
- Innovative Extension of NB: By integrating ICG and VGAE, the method enhances NB classification performance for numerical attributes, representing a novel extension of NB.
- Broader Applicability: The approach is well-suited for numerical attribute classification tasks and introduces a new perspective on instance correlation learning, making it potentially adaptable to other graph-based classification tasks.

Weaknesses:
- Unassessed Computational Cost: The construction of ICG and the training of VGAE increase computational complexity. However, the paper does not provide an analysis of runtime or computational complexity.
- Lack of Theoretical Support: While experimental results support the claims, the paper lacks a formal analysis of the independence and Gaussianity of the VGAE-generated attributes, as well as a convergence proof for attribute weighting optimization.

**Questions For Authors:**

The paper presents extensive experimental results but lacks a complete theoretical derivation, as mentioned in the weaknesses. Providing a detailed theoretical derivation would improve the overall score.

**Relation To Broader Scientific Literature:**

This paper integrates an enhanced NB approach with graph representation learning, leveraging instance correlations to optimize numerical attribute classification. Unlike traditional NB improvements that solely rely on the independence assumption, this method introduces a novel perspective by incorporating instance relationships. Moreover, it aligns with GNN-related research, bridging the gap between probabilistic classification and graph-based learning.

**Theoretical Claims:**

The theoretical claims of this paper primarily lie in the fact that the new attributes generated by ICGNB align with the core assumptions of NB, and that attribute weighting optimization enhances classification performance. However, the paper does not provide a rigorous mathematical proof; instead, these claims are validated empirically through experiments.

---

> ### Author Rebuttal · Authors · 2025-04-01
>
> Thank you very much for your valuable comments. We sincerely appreciate the time and effort you have dedicated to reviewing our work. Below, we provide detailed responses to each of your concerns.
>
> **Author Response to Computational Complexity:** We complement the time complexity analysis as follows: In Algorithm 1, line 3 constructs a full connection graph with a time complexity of $O(n^{2})$.  Lines 4-8 calculate the Euclidean distance with a time complexity of $O(n^{2}m)$. Line 9 sorts edges with a time complexity of $O(n^{2}logn)$. Lines 10-18 add edges to ICG with a time complexity of $O(n^{2}\alpha(n))$, where $O(\alpha(n))$ is the time complexity of checking if two vertices are reachable. Lines 19-20 add self-connecting edges with a time complexity of $O(n)$. Due to $m$ usually being greater than $logn$ and $\alpha(n)$, considering only the highest-order terms, the overall time complexity of Algorithm 1 is $O(n^{2}m)$. In Algorithm 2, lines 4-5 transform ICG into $G$ and initialize the parameters in VGAE with a time complexity of $O(n^2)$. Lines 6-11 train a VGAE with a time complexity of $O(P(nm^2+n^{2}m))$. Line 12 generates new attributes with a time complexity of $O(nm^2)$. Lines 13-17 augment original attributes with a time complexity of $O(nm)$. Lines 18-23 train a GNB with a time complexity of $O(knm)$. Lines 24-25 weight augmented attributes with a time complexity of $O(\beta(m))$, where $\beta(m)$ has a linear relationship with $m$.  Due to $n$ usually being greater than $m$, considering only the highest-order terms, the overall time complexity of Algorithm 2 is $O(Pn^{2}m)$. In the final version of the paper, we will add all these time complexity analyses.
>
> **Author Response to Theoretical Support:** We perform a formal analysis of Gaussianity, independence, and convergence proof for attribute weighting optimization as follows:
>
> 1. Gaussianity: VGAE ensures that the distribution of the embedding vectors closely approximates $p(\mathbf{Z})$, which is an independent zero-mean multivariate Gaussian distribution with unit variances. Each variate in $p(\mathbf{Z})$ corresponds to a new attribute in $\mathbf{Z}$, and thus the new attribute closely approximates the Gaussian distribution.
>
> 2. Independence: The independent zero-mean in $p(\mathbf{Z})$ ensures that the covariance matrix of $p(\mathbf{Z})$ is a diagonal matrix, which implies that covariances between variates (new attributes) are zero, i.e., variates (new attributes)  are independent of each other.
>
> 3. Convergence proof for attribute weighting optimization: In ICGNB, we perform the gradient descent search by using the L-BFGS algorithm. According to [1],  the L-BFGS algorithm is globally convergent on uniformly convex problems. The objective function in our optimization maximizes the conditional log-likelihood (CLL)  of the attribute weighted GNB, which is a uniformly convex problem. Therefore, the optimization process is guaranteed to converge.
>
> [1] Pearl, J. Probabilistic reasoning in intelligent systems-net-works of plausible inference. Morgan Kaufmann series in representation and reasoning. Morgan Kaufmann, 1989.
>
> **Author Response to Sensitivity Analysis:**  The hyperparameters in ICGNB include the number of iterations $P$ and the learning rate $\eta$, which are set to default values of 500 and 0.01, respectively. To address the reviewer’s concerns, we conduct parameter sensitivity experiments on the first real-world dataset (appendicitis) to evaluate ICGNB’s performance. In each experiment, we fix one hyperparameter and vary the remaining one and use the same setup described in Section 4.1. The results are as follows:
> |$P$|300|400|500|600|700|
> |:--:|:--:|:--:|:--:|:--:|:--:|
> |Average Accuracy (%)|88.64|88.18|89.09|88.64|90.00|
> ||
>
> |$\eta$|0.005|0.0075|0.01|0.0125|0.015|
> |:--:|:--:|:--:|:--:|:--:|:--:|
> |Average Accuracy (%)|88.64|90.00|89.09|89.09|88.18|
> ||
>
> These results show that ICGNB’s performance varies slightly with changes in hyperparameter values, which demonstrates that ICGNB is insensitive to parameters.
>
> **Author Response to Essential References:** Indeed, GraphSAGE is closely related to our ICGNB and provides an effective graph representation learning strategy. To address the reviewer’s concerns, we will cite GraphSAGE and discuss its correlation with our study. In addition, we conduct an experiment to explore whether GraphSAGE can be used for our ICGNB. Specifically, we replace the graph convolution function of VGAE with that of GraphSAGE and perform a stratified hold-out validation on 24 real-world datasets. The average classification accuracy is as follows:
> ||ICGNB_VGAE|ICGNB_GraphSAGE|
> |:--:|:--:|:--:|
> |Average Accuracy (%)|80.22|80.46|
> ||
>
> The results show that the graph convolution function of GraphSAGE can also be used for our ICGNB.

---

### Official Review · Reviewer_tsnb · 2025-03-14

**Overall Recommendation:** 4

**Summary:**

The paper introduces **Instance Correlation Graph-based Naïve Bayes (ICGNB)**, a novel enhancement of the **Gaussian Naïve Bayes (GNB)** classifier. Traditional Naïve Bayes methods assume conditional independence among attributes, limiting their effectiveness, especially for numerical data. The proposed **ICGNB** method addresses this by incorporating **correlations among instances**, which have been largely ignored in prior research.

**Main Algorithmic Ideas:**
1. **Instance Correlation Graph (ICG) Construction**: Instances are connected in a graph based on similarity, forming an **Instance Correlation Graph (ICG)**.
2. **Graph-Based Representation Learning**: A **Variational Graph Auto-Encoder (VGAE)** generates new attributes from the ICG, capturing latent instance correlations.
3. **Attribute Augmentation and Weighting**: The generated attributes are combined with original ones, and a weighting scheme optimizes attribute significance, reducing redundancy.

**Main Findings & Results:**
- **ICGNB outperforms traditional GNB and state-of-the-art Naïve Bayes enhancements** across multiple datasets.
- It significantly improves classification accuracy by leveraging graph-based attribute generation and augmentation.
- Empirical validation on **24 real-world datasets** and synthetic data confirms **ICGNB’s robustness and generalization ability**.

By integrating instance correlation insights into Naïve Bayes classification, ICGNB presents a **new paradigm** for improving probabilistic classifiers with graph-based learning techniques.

**Claims And Evidence:**

The paper presents **Instance Correlation Graph-based Naïve Bayes (ICGNB)** as an improvement over **Gaussian Naïve Bayes (GNB)**, claiming that leveraging instance correlations via graph-based representation learning significantly enhances classification performance. Most claims are backed by **theoretical justification and empirical evidence**, but there are some concerns regarding the **generality and robustness** of the approach.

**Supported Claims:**
1. **Effectiveness of ICGNB**: The claim that ICGNB outperforms GNB and other enhanced Naïve Bayes models is well-supported. The **Wilcoxon signed-rank test** across **24 real-world datasets** provides **statistically significant** improvements, strengthening the empirical validation.
2. **Attribute Independence and Gaussianity**: The paper convincingly shows that the **generated attributes align better with the Gaussian assumption**, validated through the **Kolmogorov-Smirnov test**.

**Potentially Problematic Claims:**
1. **Scalability**: The computational overhead of constructing the **Instance Correlation Graph (ICG)** and training the **Variational Graph Auto-Encoder (VGAE)** is not fully analyzed. Large-scale applications may face challenges.
2. **Generality**: The method is benchmarked on **numerical datasets only**. Performance on **mixed or high-dimensional categorical data** remains unclear.
3. **Causality vs. Correlation**: While ICG captures correlations, it does not necessarily improve **causal relationships**, potentially leading to **overfitting**.

**Essential References Not Discussed:**

The paper effectively cites foundational works on **Naïve Bayes improvements, graph-based learning, and variational autoencoders**, but **some key references in graph-based Bayesian learning and scalable feature learning are missing**.

1. **Graph-Based Bayesian Models**:
   - The paper does not discuss **Graphical Models for Naïve Bayes Extensions**, such as **Tree-Augmented Naïve Bayes (TAN)** (Friedman et al., 1997), which also address feature dependencies while maintaining Naïve Bayes simplicity.
   - **Relational Bayesian Models** (Getoor & Taskar, 2007) have explored **leveraging inter-instance relationships** in probabilistic classification.

2. **Scalability of Graph Autoencoders**:
   - The paper uses **Variational Graph Autoencoders (VGAE)** (Kipf & Welling, 2017) but does not discuss **scalable alternatives** such as **GraphSAGE (Hamilton et al., 2017)**, which may offer better efficiency for large-scale applications.

3. **Feature Learning for Naïve Bayes**:
   - **Autoencoder-based feature learning for probabilistic models** (Kingma & Welling, 2014) is relevant but not fully explored in terms of direct Naïve Bayes applications.

**Overall Suggestion**
Discussing **TAN, relational Bayesian models, and scalable GNN methods** would provide **a more complete context** for the proposed approach and highlight its novelty better.

**Experimental Designs Or Analyses:**

The experimental design of the paper is **generally well-structured**, incorporating a **broad set of real-world datasets**, **comparative baselines**, and **statistical validation**. However, there are some **limitations in dataset diversity and certain evaluation aspects** that could impact the validity of the findings.

 **Checked Experimental Designs and Analyses:**

1. **Dataset Selection and Preprocessing:**
   - The use of **24 numerical datasets from KEEL** ensures that the method is tested across multiple domains. However, it **excludes categorical or mixed-type data**, limiting generalizability beyond purely numerical datasets.
   - **Z-score normalization** is applied, which is **appropriate** for Gaussian-based methods but may not reflect real-world data preprocessing variations.

2. **Comparative Baselines:**
   - The paper compares ICGNB against **five well-established Naïve Bayes enhancements (WANBIA, CFWNB, AG-NBC, AE-NBC, and GNB)**, making the evaluation **comprehensive**.
   - However, the **absence of deep learning-based classifiers or ensemble methods** as additional baselines means the **broader competitiveness** of ICGNB remains unclear.

3. **Statistical Evaluation and Significance Tests:**
   - The **Wilcoxon signed-rank test** is an **appropriate choice** for measuring statistical significance in performance comparisons.
   - However, the paper does not include **confidence intervals for accuracy improvements**, which would provide a clearer measure of variability.

4. **Scalability and Computational Analysis:**
   - There is **no discussion of computational efficiency**, particularly regarding **VGAE training costs** and **graph construction overhead**.
   - The method’s viability for **large-scale datasets** is uncertain.

**Potential Issues and Suggestions:**
- The exclusion of **non-numerical datasets** limits generalizability.
- The **lack of alternative evaluation metrics** (e.g., runtime analysis, robustness to noise, or class imbalance) makes it hard to assess **real-world applicability**.
- **Computational cost analysis** is missing, which could be a key bottleneck for practical deployment.

**Overall Assessment:**
The experimental setup is **rigorous in its scope and statistical validation**, but additional **efficiency analysis, dataset diversity, and robustness tests** would enhance the study’s reliability and practical impact.

**Methods And Evaluation Criteria:**

The proposed **Instance Correlation Graph-based Naïve Bayes (ICGNB)** method is evaluated using **well-established datasets and appropriate evaluation criteria**, making the study **methodologically sound**. However, there are **certain limitations in dataset diversity and scalability analysis** that should be considered.

 **Strengths of Methods and Evaluation:**
1. **Benchmark Datasets**: The study uses **24 real-world datasets from KEEL**, covering various domains and numerical attributes. This is **appropriate for testing Gaussian Naïve Bayes (GNB) extensions**.
2. **Comparative Baselines**: The comparison against **state-of-the-art Naïve Bayes enhancements** (WANBIA, CFWNB, AG-NBC, AE-NBC, and GNB) is **comprehensive and fair**.
3. **Statistical Validation**: The **Wilcoxon signed-rank test** ensures that reported improvements are **statistically significant**, strengthening the reliability of results.

**Potential Limitations:**
1. **Dataset Scope**: The evaluation **excludes categorical or mixed-type datasets**, limiting its **generalizability** beyond numerical data.
2. **Scalability Considerations**: While VGAE improves feature representation, **its computational cost is not analyzed**. The method’s efficiency on **large-scale datasets** is uncertain.
3. **Alternative Evaluation Metrics**: The study **focuses primarily on classification accuracy**, without analyzing robustness to **adversarial noise, missing data, or class imbalance**.

 **Overall Assessment:**
The methodology is **well-structured** and **empirical validation is strong**, but additional **scalability analysis** and **broader dataset selection** would enhance the **practical applicability** of ICGNB.

**Other Comments Or Suggestions:**

No additional comments or suggestions.

**Other Strengths And Weaknesses:**

**Strengths**
1. **Novel Integration of Graph-Based Learning with Naïve Bayes**: The use of **Instance Correlation Graphs (ICG)** and **Variational Graph Autoencoders (VGAE)** to enhance Naïve Bayes classification is an **original contribution**, effectively addressing the attribute independence assumption.
2. **Strong Empirical Validation**: The **comprehensive experiments on 24 datasets**, along with **statistical significance testing (Wilcoxon signed-rank test)**, provide **credible evidence** of the method’s effectiveness.
3. **Interpretability**: Unlike deep learning-based classifiers, the method **preserves the interpretability of Naïve Bayes**, making it more suitable for explainable AI applications.

**Weaknesses**
1. **Scalability Concerns**: **No discussion on computational complexity**, particularly regarding **graph construction and VGAE training**, which may hinder practical deployment on large datasets.
2. **Limited Generalization**: The approach is **only evaluated on numerical datasets**, excluding categorical and mixed-type data, limiting its **applicability in broader domains**.
3. **Clarity of Theoretical Justifications**: While empirically validated, **some theoretical claims (e.g., feature independence in VGAE-generated attributes)** lack formal proofs.

**Overall**
The paper presents an **innovative and promising approach**, but addressing **scalability and broader dataset applicability** would enhance its impact.

**Questions For Authors:**

1. **Scalability and Computational Complexity**
   - What is the computational complexity of constructing the **Instance Correlation Graph (ICG)** and training the **Variational Graph Autoencoder (VGAE)**? Have you evaluated runtime performance on larger datasets?
   - A response showing that ICGNB is computationally efficient would **strengthen confidence in its practicality**.

2. **Applicability to Categorical and Mixed-Type Data**
   - Since the experiments focus only on **numerical datasets**, how would ICGNB handle **categorical or mixed-type features**? Would discretization or embedding techniques be required?
   - If the method is adaptable to broader data types, its **generalizability and impact would increase**.

3. **Independence and Gaussianity of Generated Attributes**
   - The empirical results suggest that **VGAE-generated attributes align better with the Naïve Bayes assumptions**, but is there a **theoretical justification** for this claim?
   - A formal proof or additional justification would **strengthen the theoretical contributions of the paper**.

**Relation To Broader Scientific Literature:**

The key contributions of this paper are closely related to several existing lines of research in **Naïve Bayes improvements, graph-based learning, and representation learning with autoencoders**. While the paper introduces a novel **Instance Correlation Graph-based Naïve Bayes (ICGNB)** framework, its ideas build upon well-established methodologies in **probabilistic classification and graph neural networks**.

**Relation to Prior Research:**

1. **Naïve Bayes Enhancements:**
   - The paper follows a long tradition of **improving Naïve Bayes by addressing its independence assumption**. Previous works have explored **attribute weighting (WANBIA, CFWNB)** and **attribute transformation (AG-NBC, AE-NBC)**.
   - ICGNB extends these by **introducing instance-instance correlations**, which are generally ignored in prior Naïve Bayes improvements.

2. **Graph-Based Representation Learning:**
   - The use of **Instance Correlation Graphs (ICG)** aligns with advances in **graph-based semi-supervised learning**, particularly **Graph Convolutional Networks (GCN)** and **Variational Graph Autoencoders (VGAE)** (Kipf & Welling, 2017).
   - VGAE has been widely used for **latent feature extraction**, and this paper leverages it to **generate new attributes** that improve classification performance.

3. **Attribute Augmentation and Weighting:**
   - The method shares similarities with **feature learning techniques in deep learning**, where new attributes are derived via **embedding-based transformations**.
   - However, unlike deep learning methods, ICGNB retains the **probabilistic interpretability of GNB**.

 **Broader Impact and Limitations:**
- The **combination of Naïve Bayes with graph-based learning is novel**, but similar ideas have been explored in **relational learning and Bayesian networks**.
- The paper focuses on **numerical data only**, which limits its applicability in domains where **categorical attributes** play a significant role.
- While the approach improves Naïve Bayes, it does not explore how it compares to **modern deep learning classifiers**.

**Overall Assessment:**
ICGNB contributes meaningfully to **graph-enhanced probabilistic classification**, bridging **Naïve Bayes improvements with representation learning**. However, its relation to broader **graph-based Bayesian models and deep learning-based feature learning** could be explored further.

**Theoretical Claims:**

The paper presents **theoretical claims** primarily related to **the effectiveness of the Instance Correlation Graph (ICG), the variational graph auto-encoder (VGAE) for attribute generation, and the weighted Gaussian Naïve Bayes (GNB) formulation**. While the overall framework is well-motivated, some theoretical aspects require **closer scrutiny**.

**Checked Theoretical Claims:**
1. **ICG Construction and Sparsity**:
   - The paper claims that **ICG captures meaningful instance correlations** while maintaining **a sparse structure**. The **edge selection algorithm** ensures connectivity using a minimal number of edges. This is **conceptually sound**, though a formal proof of optimality is missing.

2. **VGAE-Based Attribute Generation**:
   - The theoretical justification relies on **graph convolutional embedding** preserving instance relationships. The **variational objective (ELBO) is correctly formulated** following standard **VGAE methodology** (Kingma & Welling, 2014). However, the assumption that **generated attributes are inherently more independent** is **not formally proven**, only empirically suggested.

3. **Attribute Weighting via Conditional Log-Likelihood (CLL) Maximization**:
   - The gradient-based weight optimization process follows a **standard likelihood maximization framework**. However, convergence guarantees or sensitivity analysis are **not provided**.

**Potential Issues:**
- **No proof of Gaussianity**: The claim that **VGAE-generated attributes better fit Gaussian assumptions** is only empirically tested, lacking **a theoretical derivation**.
- **Scalability of VGAE training**: No theoretical analysis of **complexity bounds** is presented.

**Overall Assessment:**
The proofs presented are **generally correct**, but some **key claims (e.g., independence of generated features, scalability) lack formal justification** and are **only validated empirically**. A more rigorous **theoretical treatment** of these aspects would strengthen the paper.

---

### Decision · Program_Chairs · 2025-05-01

**Decision:**

Accept (spotlight poster)

**Comment:**

This paper proposes an enhancement of Gaussian Naive Bayes method called ICGNB. Its advantages include: 1) combining graph-based learning with probabilistic modeling and introducing a new angle to Naive Bayes by modeling inter-instance relationships; 2) maintaining the clarity of GNB while enhancing performance; 3) providing formal arguments on Gaussianity, independence, and convergence of attribute weighting. However, there are concerns about scalability, generalization to data types, and a lack of comparison with broader models. Overall, this is a qualified paper and I recommend acceptance.